# Personalized care planning and shared decision making in collaborative care programs for depression and anxiety disorders: A systematic review

Matthew Menear [1,2]*, Ariane Girard[1,2], Michèle Dugas[1], Michel Gervais[3], Michel Gilbert[4], Marie-Pierre Gagnon[1,5]

1 VITAM Research Centre for Sustainable Health, Quebec, Quebec, Canada, 2 Department of Family Medicine and Emergency Medicine, Université Laval, Quebec, Quebec, Canada, 3 Centre Intégré Universitaire de Santé et de Services Sociaux de la Capitale-Nationale, Quebec, Quebec, Canada, 4 Centre National d'Excellence en Santé Mentale, Quebec, Quebec, Canada, 5 Faculty of Nursing, Université Laval, Quebec, Quebec, Canada

* matthew.menear.1@ulaval.ca

**Data Availability Statement:** Relevant data are provided within the manuscript and its Supporting Information files. The minimal data set for this

## Abstract

### Background

Collaborative care is an evidence-based approach to improving outcomes for common mental disorders in primary care. Efforts are underway to broadly implement the collaborative care model, yet the extent to which this model promotes person-centered mental health care has been little studied. The aim of this study was to describe practices related to two patient and family engagement strategies–personalized care planning and shared decision making–within collaborative care programs for depression and anxiety disorders in primary care.

### Methods

We conducted an update of a 2012 Cochrane review, which involved searches in Cochrane CCDAN and CINAHL databases, complemented by additional database, trial registry, and cluster searches. We included programs evaluated in a clinical trials targeting adults or youth diagnosed with depressive or anxiety disorders, as well as sibling reports related to these trials. Pairs of reviewers working independently selected the studies and data extraction for engagement strategies was guided by a codebook. We used narrative synthesis to report on findings.

### Results

In total, 150 collaborative care programs were analyzed. The synthesis showed that personalized care planning or shared decision making were practiced in fewer than half of programs. Practices related to personalized care planning, and to a lesser extent shared decision making, involved multiple members of the collaborative care team, with care managers playing a pivotal role in supporting patient and family engagement. Opportunities for

study has been published in the Harvard Dataverse and is available using the following URL: https://doi.org/10.7910/DVN/MWZOIE.

**Funding:** This work was funded by a Canadian Institutes for Health Research (CIHR) knowledge synthesis grant (number 201505KRS-350972-KRS-CFBA; https://cihr-irsc.gc.ca). Dr. Menear was supported by a Junior 1 salary award from the Fonds de recherche du Québec - Santé (FRQS). The funding sources for this research had no role in the study design, in the collection, analysis or interpretation of data, in the writing of the report, or the decision to submit the article for publication.

**Competing interests:** The authors have declared that no competing interests exist.

quality improvement were identified, including fostering greater patient involvement in collaborative goal setting and integrating training and decision aids to promote shared decision making.

## Conclusion

This review suggests that personalized care planning and shared decision making could be more fully integrated within collaborative care programs for depression and anxiety disorders. Their absence in some programs is a missed opportunity to spread person-centered mental health practices in primary care.

## Introduction

Over the past two decades, collaborative care has attracted worldwide interest as an effective, team-based approach to managing common mental disorders in primary care [1–4]. The collaborative care model aims to promote greater mutual support between primary care and mental health providers and the delivery of more coordinated, integrated, and evidence-based mental health services. While the earliest collaborative care programs emphasized closer relationships between family physicians and psychiatrists [5, 6], the model has evolved over the past two decades to include a broader range of professionals that can collaborate in care [7]. Views on the roles of patients and families as active partners in collaborative care have similarly evolved [8, 9]. There has been a growing acknowledgments that much of the work to promote patient recovery happens outside the clinic, and involves patients and families becoming better informed about their conditions, taking greater responsibility in illness management and learning to adopt psychosocial and lifestyle changes that can improve their well-being (9). Today, some view patients and families as the *most* important members of the collaborative care team and emphasize the need for supports in helping them become partners in care [10].

As the literature on patient engagement in healthcare evolves, it is becoming clear that there are numerous ways for patients and families to be engaged in mental health care [11]. However, some practices may reflect a more meaningful partnership with collaborative care teams. For instance, in the early stages of care, providers would work collaboratively with patients to establish a care plan that integrates their goals for treatment and recovery. When care planning adopts a personalized approach, patients are engaged in open-ended discussions about the problems that matter most to them and how to align care to their unique needs and priorities [12]. Patients are encouraged to define recovery goals not only specific to their symptomatology, but to other spheres of their life that contribute to their well-being (e.g. behavior changes, development of new skills, social roles or activities to pursue, personal challenges to overcome) [12–14]. For some patients, the presence of family members in these discussions may be essential. Such discussions would be followed by a jointly developed action plan that identifies how patients, families, and providers will work together to achieve mutually agreed upon goals and outcomes. Documenting this plan is an important step as it provides a central record that patients and members of the care team can routinely review to track progress in the achievement of goals and outcomes throughout the course of care [12–14]. The development of individually tailored care plans that consider patient preferences is not a new standard of practice; it has been considered a best practice for common mental disorders since the 1990s [15–17].

Another critical element of care planning and the overall management of mental disorders is shared decision making (SDM) [13, 18]. SDM is an interpersonal process by which health

decisions are deliberated upon and made jointly by patients and their care providers taking into consideration the best available evidence, clinical judgment, and patients' preferences [19]. This process ensures that patients make informed decisions based on an understanding of the range of options available to them, the benefits and risks of these options, and how their own personal preferences and values align with each option. People with common mental disorders may encounter a number of decisions over the course of their care that are preference-sensitive and that would warrant an SDM approach [18, 20]. They may also hold strong preferences for some options over others, such as the types of treatments they wish to receive. Family members may also desire to support their loved ones and participate in the decision-making process [21]. Similar to personalized care planning, principles of SDM have long been considered good clinical practice and were promoted in some of the earliest collaborative mental health care programs [22, 23].

Whether personalized care planning and SDM are consistently emphasized and practiced within collaborative mental health care is an important issue. Common disorders such as depressive and anxiety disorders impose a massive burden worldwide [24] and collaborative care is widely recognized as an effective model of care that can achieve positive impacts on population health [9, 25, 26]. The past decade has seen growing support for the dissemination and scaling-up of collaborative care [25, 27–29] and such models are becoming a central component of national mental health policies [27, 30]. Yet, there have been few investigations into the person-centeredness of collaborative care programs and the extent to which patients and families are actively engaged as partners in care. If this model is to be a standard for the delivery of evidence-based care for common mental disorders in primary care, it is imperative that these programs be as person-centered as possible. The overall aim of this study was thus to describe how two key engagement strategies—personalized care planning and SDM–have been practiced within collaborative care programs for common mental disorders. We formulated the following research questions:

1. How often are personalized care planning and SDM strategies included within collaborative care programs for common mental disorders?

2. Who participates in personalized care planning and SDM?

3. How and when is personalized care planning and SDM practiced?

## Methods

We conducted a systematic review examining strategies for engaging patients and families in collaborative care programs for depressive and anxiety disorders. In a previous article, we reported the full range of engagement strategies that had been used in these programs [11]. In the current article, we report the more detailed findings related to care planning and SDM. The protocol for the overall review was registered with PROSPERO (www.crd.york.ac.uk/prospero), number CRD42015025522. We follow PRISMA guidelines in reporting our methods and results [31].

### Search strategy

Our search strategy was designed by an information specialist to retrieve two types of articles: a) reports of clinical trials of collaborative care programs, and b) 'sibling' reports, i.e. clinical trial protocols, quantitative or qualitative sub-studies, or other reports linked to these programs. To identify clinical trial articles, we performed an update of the 2012 Cochrane systematic review of collaborative care interventions for depression and anxiety disorders. We

replicated the search strategies used by the review authors, which included keyword searches for depression, anxiety, and collaborative care in the Cochrane Collaboration Depression, Anxiety and Neurosis Group (CCDAN) registers and CINAHL database. The CCDAN registers contain trials and references related to depression, anxiety and neurosis drawn from weekly systematic searches in MEDLINE, EMBASE, PsycINFO, and Cochrane CENTRAL [1]. The CCDAN registers were initially searched from January 2011 until June 2016. The CINAHL database was searched from January 2009 until June 2016 (see all search strategies in S1 File). To identify more recent or ongoing collaborative care trials, we searched three trial registers (ClinicalTrials.gov, WHO ICTRP, EU Clinical Trials Register) using a simplified set of search terms (e.g. 'collaborative care', 'integrated care', 'stepped care', 'case management'). We also created monthly email alerts to monitor the publication of new articles on collaborative care programs for depression and anxiety disorders in the MEDLINE, Embase, and PsycINFO databases based on our initial detailed search strategy. The monthly alerts were active since June 2016, with the last verification of the literature being completed in June 2020.

For all clinical trials of collaborative care, we identified sibling articles related to the trial using 'cluster searching' [32], which included reference list searches, author searches in Web of Science, reverse citation searches, and searches in Google Scholar using study trial names (e.g. IMPACT study). The purpose of identifying sibling articles was to retrieve additional details of programs and engagement strategies (including care planning and SDM) that were not included in the primary trial articles for collaborative care programs.

### Eligibility criteria

Clinical trial studies were included if they met the following criteria. First, participants were individuals of any age with a primary diagnosis of depression or anxiety disorder. Second, interventions were consistent with the collaborative care model and featured: (1) a multidisciplinary approach to care involving at least one primary care practitioner and another health professional; (2) a structured management plan (e.g. use of guidelines or algorithms, evidence-based treatments); (3) a systematic approach to patient follow-up; and (4) mechanisms for enhanced communication between providers (e.g. interdisciplinary team meetings, clinical supervision). Third, studies had to be randomized controlled trials or clinical controlled trials that included an eligible comparator, such as usual care or an alternative collaborative care intervention. Fourth, studies also had to report at least one outcome related to changes in depression or anxiety status, medication use, quality of life, or satisfaction with care. Sibling reports were eligible for inclusion if they described or provided additional empirical data on a collaborative care program deemed eligible for the review.

### Selection process

A team of five researchers and research professionals independently conducted the initial title-abstract screening. Two review authors then independently reviewed the full-texts of all relevant articles. For both initial and full-text screening, disagreements between review authors were resolved through discussion and, if necessary, by consulting the primary author (MM).

### Data extraction

We extracted the following data into a piloted, structured Excel form: a) study characteristics, b) collaborative care program characteristics, c) participant characteristics, and d) presence and characteristics of personalized care planning and SDM strategies (e.g. types of professionals or other actors involved, care plan and SDM processes and components, duration/timing of these processes, evaluations of these processes). We considered personalized care planning

to be present when: a) in the descriptions of the intervention there was some reference to a care plan or treatment plan; b) care planning was *personalized*, i.e. there was clear reference to patients being involved in developing the care plan or acting as partners in other components of the care planning process; and c) at least one component of the care planning process was clearly described. We used the framework proposed in Coulter et al.'s Cochrane review [13] to define seven components of personalized care planning, which were organized into two main phases: 1) Developing the care plan (i.e. Preparation, Goal setting, Action planning, and Documentation), and 2) Coordinating care and adjusting the plan (i.e. Coordination, Supporting, and Reviewing). We further included 'Relapse prevention planning' as a fourth component in the latter phase, as this was found to be a distinct process in several programs.

For data on SDM, we assessed descriptions of the decision-making process used with patients and considered a process to minimally reflect SDM when at least two of the following processes [33] were reported: a) the provider identified a decision to be made or explained the problem requiring a decision, b) the provider and patient exchanged information related to a decision, c) the provider presented options to the patient, d) the preferences of the patient were explored, e) the pros and cons of each option were discussed, f) the provider and patient made a decision together, and g) the provider and patient revisited a particular decision. Examples of data extracted to describe processes related to personalized care planning and SDM are provided in S2 File.

For each program, data was first extracted from the primary trial article and then additional details were sought from the sibling articles until data saturation was achieved. A single review team member performed the extraction, supported by a codebook providing definitions and examples for collaborative care programs and care planning and SDM strategies. The primary author performed a full verification of the accuracy of all data extracted.

## Data analysis and synthesis

Given the descriptive nature of our objectives and high degree of heterogeneity across studies, we adopted a narrative approach to data synthesis. Results related to the presence and characteristics of personalized care planning and SDM strategies were summarized using descriptive statistics as well as tables with text extractions describing each strategy. We compared and contrasted the study, program and participant characteristics of programs that included care planning and SDM against programs that did not include these strategies and explored patterns both across studies and within studies featuring each strategy.

## Results

### Search results

The 2012 Cochrane review identified 79 unique collaborative care programs and 317 'sibling' reports that described study protocols or sub-studies related to these trials [1]. Searches for the review update yielded 4643 records after removal of duplicates. Screening of titles and abstracts led to the exclusion of 4339 records, leaving 304 records eligible for full-text screening. Following full-text screening, we excluded another 239 records, leaving a total of 65 trial articles that described 51 unique collaborative care programs. We also identified 20 additional programs through the trial registry searches and monthly email alerts that, together with the programs identified in the 2012 Cochrane review, produced an overall total of 150 unique collaborative care programs for depressive or anxiety disorders. Using cluster searching, we retrieved an additional 130 sibling reports describing these programs, for a total of 447 sibling reports overall (Fig 1).

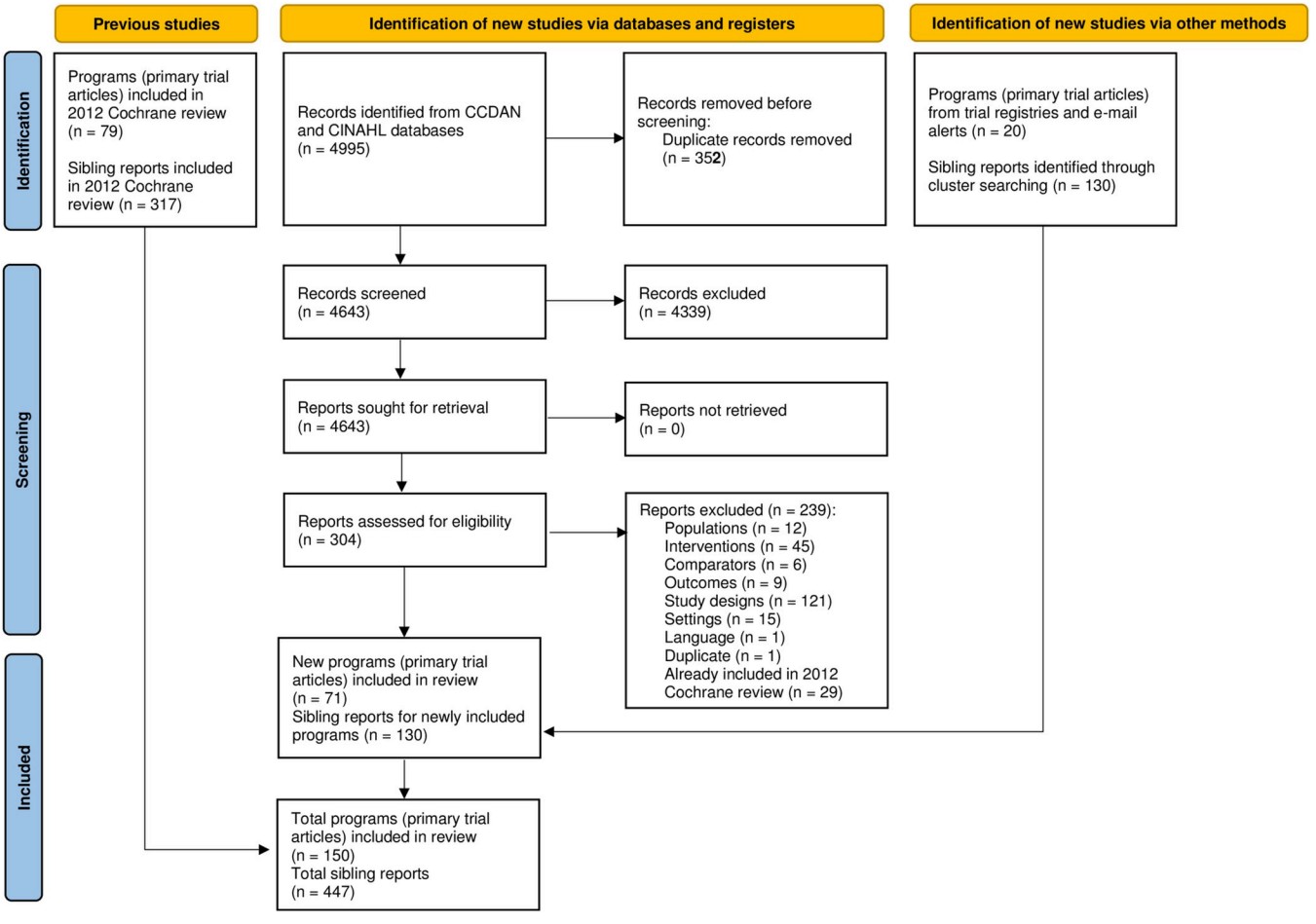

**Fig 1. PRISMA flow diagram.**

## Programs featuring personalized care planning and shared decision making

Among the 150 collaborative care programs, 63 programs (42%) included either personalized care planning or SDM strategies. There were 51 programs (34% of all programs) that featured personalized care planning and 43 programs (29% of all programs) that featured SDM. Of the 63 programs with either personalized care planning or SDM, 31 programs (49%) featured both strategies, 20 programs (32%) featured only personalized care planning, and 12 programs (19%) featured only SDM. There were no discernable patterns or differences with respect to study characteristics (e.g. publication dates, study location), program characteristics (e.g. type of collaborative care intervention), or participant characteristics (e.g. primary diagnoses, age, gender) when comparing studies that did vs. did not feature personalized care planning or SDM as a part of the program. While there was an overall lower proportion of programs evaluated during the 1990s that featured these strategies (1/6 programs, 17%), we did not observe that a significantly higher proportion of programs conducted between 2010–2019 adopted these strategies (25/54 programs, 46%) compared to programs implemented in the years 2000–2009 (38/90 programs, 42%). There was however a relatively higher proportion of programs adopting personalized care planning or SDM within studies targeting children or adolescents (3/4 programs, 75%) compared to studies that targeted adults (52/126 programs, 41%), only

older adults (7/16 programs, 44%) or mixed youth and adult populations (1/4 programs, 25%). The characteristics for the 63 programs that featured personalized care planning or SDM are summarized in Table 1.

## Participants in personalized care planning

When personalized care planning was featured within a program (N = 51 programs), the process frequently involved multiple professionals within the team (Table 2). Primary care physicians (88% of programs) and care managers (98% of programs) were the two types of providers most often involved. The role of care manager was held by a range of professionals, most often nurses, social workers or psychologists. Psychiatrists also contributed to care planning in 69% of programs. Finally, in 43% of programs, other professionals such as physician specialists (e.g. cardiologists), psychologists, pharmacists, nurses and social workers (not in a

**Table 1. Characteristics of programs included in the review.**

| Program | Setting and Country | Population | Intervention | Presence of PCP and/or SDM |
|---|---|---|---|---|
| Asarnow 2005 [34] Youth Partners in Care | Primary care services in two states in the U.S. | Youth (13–21 years) with depression | Collaborative care | PCP and SDM |
| Barley 2014 [35] UPBEAT | Primary care clinics in South London UK | Adults (18+) with depression and coronary heart disease | Nurse-based collaborative care | PCP only |
| Battersby 2013 [36] Flinders Program | Primary care services in Adelaide, Australia | Adult veterans (18+ years) with common mental disorders, alcohol misuse and chronic medical conditions | Integrated collaborative care for mental and alcohol use disorders | PCP and SDM |
| Bjorkelund 2018 [37] PRIM-CARE | Primary care services in Västra Götaland region, Sweden | Adults (18+ years) with depression | Stepped collaborative care | PCP and SDM |
| Blanchard 1995 [38] Gospel Oak Depression Study | Primary care services in England, U.K. | Older adults (60+ years) with depression | Nurse-based collaborative care | PCP only |
| Bosanquet 2017 [39] CASPER Plus | Primary care services in England, U.K. | Older adults (65+ years) with depression | Collaborative care | SDM only |
| Buszewicz 2010 [40] ProCEED | Primary care services in England, Northern Ireland, and Scotland, U.K. | Adults (18+ years) with chronic depression | Nurse-based collaborative care | PCP and SDM |
| Carney 2016 [41] | Outpatient cardiology services in two states in the U.S. | Adults (18+ years) with depression and coronary heart disease | Collaborative care | PCP and SDM |
| Chaney 2011 [42] TIDES | Primary care services in five states in the U.S. | Adults (18+ years) with depression | Collaborative care | PCP and SDM |
| Clarke 2005 [43] | Pediatric primary care services in Oregon, U.S. | Youth (12–18 years) with depression | CBT-based collaborative care | PCP and SDM |
| Cooper 2013 [44] BRIDGE | Primary care clinics in two states in the U.S. | Adults (18–75 years) with depression | Patient-centered, culturally-tailored collaborative care | SDM only |
| Coventry 2015 [45] COINCIDE | Primary care services in England, U.K. | Adults (18+ years) with depression and diabetes and/or cardiovascular disease | Integrated collaborative care for depression and diabetes or cardiovascular disease | PCP and SDM |
| Davidson 2013 [46] CODIACS | Hospital settings in five states in the U.S. | Adults (35+ years) with depression and acute coronary syndrome | Stepped collaborative care | PCP and SDM |
| Dietrich 2004 [47] RESPECT-Depression | Primary care services in several states in the U.S. | Adults (18+ years) with depressive disorders | Telephone-based collaborative care | SDM only |
| Dwight-Johnson 2005 [48] | Oncology services in California, U.S. | Latino women (18+ years) with depression and cancer | Integrated collaborative care for depression and cancer | PCP and SDM |
| Dwight-Johnson 2010 [49] | Primary care services in California, U.S. | Latino adults (18+ years) with depression | Collaborative care | SDM only |
| Dwight-Johnson 2011 [50] | Primary care services in Washington state, U.S. | Latino adults (18+ years) with depression | Telephone and CBT-based collaborative care | SDM only |

*(Continued)*

**Table 1.** (*Continued*)

| Program | Setting and Country | Population | Intervention | Presence of PCP and/or SDM |
|---|---|---|---|---|
| Ell 2007 [51] HOPE-D | Home care services in the U.S. | Older adults (65+ years) with depression | Home-based collaborative care | PCP and SDM |
| Ell 2008 [52] ADAPt-C | Community-based services in California, U.S. | Adults (18+ years) with depression | Integrated collaborative care for depression and cancer | PCP and SDM |
| Ell 2010 [53] MDDP | Primary care services in California, U.S. | Adults (18+ years) with depression | Integrated collaborative care for depression and diabetes | PCP and SDM |
| Engel 2016 [54] STEPS UP | Primary care services in five states in the U.S. | Adult military personnel (18+ years) with depression and PTSD | Nurse and Web-based collaborative care | PCP and SDM |
| Fortney 2007 [55] TEAM | Community-based services for veterans health in three states in the U.S. | Adults (18+ years) with depression | Stepped, telemedicine-based collaborative care | PCP only |
| Grote 2015 [56] MOMCare | Public health centers in Washington state, U.S. | Pregnant women (18+ years) with depression | Stepped collaborative care | PCP and SDM |
| Hedrick 2003 [57] | Veterans Affairs primary care services in Washington state, U.S. | Adults (18+ years) with depression | Collaborative care | PCP and SDM |
| Huffman 2011 [58] SUCCEED | Hospital cardiac units in Massachusetts, U.S. | Adults (18+ years) with depression and cardiovascular disorders | Integrated collaborative care for depression and cardiovascular diseases | PCP and SDM |
| Huffman 2014 [59] MOSAIC | Cardiac units in a hospital in Massachusetts, U.S. | Adults (18+ years) with depression and/or anxiety disorders and cardiovascular disease | Stepped collaborative care | PCP only |
| Huijbregts 2013 [60] | Primary care services in several regions of the Netherlands | Adults (18+ years) with depression | Collaborative care | PCP and SDM |
| Hunkeler 2000 [61] | Primary care services in two states in the U.S. | Adults (18+ years) with depression | Telephone-based collaborative care | PCP only |
| Johnson 2014 [62] TEAMcare-PCN | Primary care services in Alberta, Canada | Adults (18+ years) with depression and diabetes | Integrated collaborative care for depression and diabetes | PCP and SDM |
| Katon 2004 [63] Pathways | Primary care services in Washington state, U.S. | Adults (18+ years) with depression | Stepped collaborative care | PCP and SDM |
| Katon 2010 [64] TEAMcare | Primary care services in Washington state, U.S. | Adults (18+ years) with depression and diabetes and/or coronary heart disease | Integrated collaborative care for depression and chronic diseases | PCP only |
| Kilbourne 2014 [65] LGCC SM Life Goals | Veteran Affairs mental health services in one state in the U.S. | Adults (18+ years) with chronic mental disorders | Collaborative care | PCP only |
| Kwong 2013 [66] | Primary care services in New York, U.S. | Chinese adults (18+ years) with depression | Collaborative care | SDM only |
| Ludman 2007 [67] | Primary care services in North Carolina, U.S. | Adults (18+ years) with depression | Telephone and CCM-based collaborative care | PCP only |
| McCusker 2008 [68] Project Direct | Primary care services in Quebec, Canada | Older adults (60+) with depression | Collaborative care | PCP only |
| McSweeney 2012 [69] | Seniors residences in Melbourne, Australia | Older adults (60+ years) with depression and dementia | Collaborative care | PCP only |
| Melville 2014 [70] DAWN | Obstetrics and gynecology services in Washington state, U.S. | Women (18+ years) with depression | Collaborative care | PCP and SDM |
| Meredith 2016 [71] VISTA | Federally Qualified Health Centers in two states in the U.S. | Adults (18–65 years) with PTSD | Collaborative care | PCP only |
| Morgan 2013 [72] | Primary care clinics in Victoria, Australia | Adults (18+ years) with depression | Nurse-based stepped collaborative care | PCP only |
| Muntigh 2013 [73] | Primary care services in Leiden, Netherlands | Adults (18+ years) with anxiety disorders | Stepped collaborative care | PCP only |
| Oslin 2003 [74] | Primary care services in Pensylvania, U.S. | Adults (18+ years) with depression | Telephone-based collaborative care | PCP only |

(*Continued*)

**Table 1.** (Continued)

| Program | Setting and Country | Population | Intervention | Presence of PCP and/or SDM |
|---|---|---|---|---|
| Pyne 2011 [75] HITIDES | Veteran Affairs HIV treatment services in three states in the U.S. | Adults (18+ years) with depression and HIV infection | Stepped, Integrated collaborative care for depression and HIV | PCP and SDM |
| Richards 2013 [76] CADET | Primary care services in England, U.K. | Adults (18+ years) with depression | Collaborative care | PCP and SDM |
| Richardson 2014 [77] ROAD | Primary care services in Washington state, U.S. | Youth (13–17 years) with depression | Collaborative care | SDM only |
| Rollman 2005 [78] | Primary care services in Pensylvania, U.S. | Adults (18–64 years) with anxiety disorders | Telephone-based collaborative care | PCP and SDM |
| Rollman 2009 [79] Bypassing the Blues | University- and community-based hospitals in Pensylvania, U.S. | Adults (18+ years) with depression in recovery from coronary artery bypass graft surgery | Telephone-based collaborative care | PCP and SDM |
| Rollman 2017 [80] | Primary care services in Pennsylvania, U.S. | Adults (18+ years) with anxiety disorders | Telephone-based stepped collaborative care | PCP and SDM |
| Rollman 2018 [81] | Primary care services in Pennsylvania, U.S. | Adults (18+ years) with depression and/or anxiety disorders | Web-based collaborative care | PCP only |
| Rost 2001 [82] | Primary care services in several states in the U.S. | Adults (18+ years) with depression | Collaborative care | SDM only |
| Simon 2004 [83] | Primary care services in Washington state, U.S. | Adults (18+ years) with depression | Telephone and CBT-based collaborative care | PCP and SDM |
| Smit 2006 [84] DRP Program | Primary care services in Groningen, Netherlands | Adults (18–70 years) with depressive disorders | Collaborative care | PCP only |
| Swindle 2003 [85] | Primary care services in Indiana, U.S. | Adults (18+ years) with depression | Nurse-based collaborative care | PCP only |
| Unutzer 2002 [86] IMPACT | Primary care services in five states in the U.S. | Older adults (60+ years) with depression | Stepped collaborative care | PCP and SDM |
| Van Beljouw 2015 [87] Lust for Life | Primary care and home care services in three regions in the Netherlands | Older adults (65+ years) with depression | Stepped collaborative care | SDM only |
| Vlasveld 2012 [88] | Occupational health services in the Netherlands | Adults (18+ years) with depression | Collaborative care | PCP and SDM |
| Weinreb 2016 [89] | Primary care services in New York, U.S. | Homeless women (18+ years) with depression | Collaborative care | PCP only |
| Wells 2000 [22] Partners in Care | Primary care services in five states in the U.S. | Adults (18+ years) with depression | Collaborative care | PCP and SDM |
| Wisner 2017 [90] | Home-based services and a women's hospital in Pennsylvania, U.S. | Women (18+ years) with postpartum depression | Telephone-based collaborative care | SDM only |
| Yawn 2012 [91] TRIPPD | Primary care practices in 21 states in the U.S. | Women (18+ years) with postpartum depression | Collaborative care | SDM only |
| Zatzick 2001 [92] | Trauma center in North Carolina, U.S. | Adolescent and adult (14–65 years) trauma survivors with depression | Collaborative care | PCP only |
| Zatzick 2004 [93] | Trauma center in Washington state, U.S. | Adults (18+ years) with depression and PTSD | Collaborative care | PCP only |
| Zatzick 2013 [94] | Primary care and trauma center services in Washington state, U.S. | Adult (18+ years) trauma survivors with PTSD | Stepped collaborative care | SDM only |
| Zimmerman 2014 [95] SMADS | Primary care services in Hamburg, Germany | Adults (18+ years) with depression and/or anxiety disorders | Nurse-based collaborative care | PCP and SDM |

PCP: Personalized care planning; SDM: Shared decision making

**Table 2. Participants in personalized care planning (PCP) and shared decision making (SDM).**

| | PCP | | SDM | |
|---|---|---|---|---|
| | (N = 51) | | (N = 43) | |
| **Types of providers involved** | **N** | **%** | **N** | **%** |
| Primary care physicians | 44 | 88 | 26 | 60 |
| Psychiatrists | 35 | 69 | 12 | 29 |
| Care managers[1] | 50 | 98 | 36 | 84 |
| Nurses | 29 | 57 | 18 | 42 |
| Social workers | 12 | 24 | 13 | 30 |
| Psychologists | 7 | 14 | 2 | 5 |
| Other mental health professionals | 6 | 12 | 5 | 12 |
| Other professionals[2] | 3 | 6 | 2 | 5 |
| Other professionals | 22 | 43 | 9 | 21 |
| Nurses | 2 | 4 | 1 | 2 |
| Social workers | 2 | 4 | 1 | 2 |
| Psychologists | 9 | 18 | 5 | 12 |
| Other mental health professionals | 2 | 4 | 0 | 0 |
| Pharmacists | 2 | 4 | 0 | 0 |
| Other specialists | 9 | 18 | 4 | 10 |
| **Number of providers involved** | | | | |
| One | 3 | 6 | 21 | 49 |
| Two | 10 | 20 | 10 | 23 |
| Three | 20 | 39 | 5 | 12 |
| Four | 16 | 31 | 6 | 14 |
| Five | 2 | 4 | 1 | 2 |
| Median | 3 | | 2 | |
| **Patients–Age groups** | | | | |
| Children or adolescents | 2 | 4 | 3 | 7 |
| Adults (18 years and above) | 42 | 84 | 36 | 84 |
| Older adults only (60 years and above) | 5 | 10 | 4 | 9 |
| Mixed youth and adults | 1 | 2 | 0 | 0 |
| **Patients–Primary diagnoses** | | | | |
| Depression | 40 | 78 | 36 | 84 |
| Anxiety | 4 | 8 | 3 | 7 |
| Depression and/or anxiety | 7 | 14 | 4 | 9 |
| **Family involvement** | | | | |
| Yes | 8 | 16 | 6 | 14 |
| No | 43 | 84 | 37 | 86 |

[1] In some programs, more than one type of professional acted as a care manager.

[2] Other professionals that acted as care managers included occupational physicians, nurse practitioners and psychotherapists.

care manager role) were involved in developing and supporting the care plan. Programs described the participation of three or more professionals in care planning in 74% of programs featuring this strategy.

Patients engaged in personalized care planning as part of collaborative care were most often adults with a primary diagnosis of major depression (Table 2). A smaller number of programs delivering collaborative care to patients with anxiety disorders (N = 11) reported engaging

patients in personalized care planning. Similarly, youth or older adult populations were engaged in personalized care planning in only 2 programs [34, 43] and 5 programs [23, 38, 51, 68, 69], respectively. Finally, we found evidence for family involvement in only 8 of the 51 programs that featured this engagement strategy [34, 43, 48, 51, 53, 96].

## Participants in shared decision making

Whereas for personalized care planning the involvement of multiple professionals was the norm, in 48% of programs patients were engaged in SDM with only a single member of the collaborative care team, most often either the care manager or primary care physician (Table 2). Care managers supported SDM in 84% of programs featuring this strategy, including in 71% of cases when only a single professional was involved. Primary care physicians participated in SDM in 60% of programs, and in 5 programs were the sole supporter of SDM. Relative to personalized care planning, psychiatrists and other professionals were less frequently involved (in 28% and 21% of programs, respectively). The involvement of three or more professionals in SDM processes was identified in 28% of programs featuring this strategy.

The profile of patients engaged in SDM resembled very closely the characteristics of patients involved in personalized care planning, with the majority of programs focusing on adults with major depression. There was also a low rate of involvement of family members in SDM (6/43 programs) [34, 48, 51, 53, 77, 96].

## Personalized care planning processes

In most programs, personalized care planning was a multi-phased process that involved different members of the collaborative care team at different points in time. The first phase is related to the development of the care plan. This involved preparation, observed in 94% of programs featuring care planning. This comprised an assessment and exchange of information with the patient. In 76% of the programs, this preparation largely took place during an initial session alone with the care manager. During this initial session, the care manager would typically assess patients' current situation and symptoms, review their history, provide education or educational materials, discuss treatment options, and explore their treatment preferences. In some programs, care managers also sought to understand patients' beliefs or explanatory models around depression or anxiety, explore potential barriers to care, or begin developing a therapeutic alliance with patients. In other programs there was no clear initial meeting between the patient and care manager but rather regular multidisciplinary team meetings in which the patient cases were reviewed and discussed between multiple professionals. In some cases, such as in the Gospel Oak program [38] and program led by Hedrick [57, 97], patients did not participate in the meetings and their involvement in the creation of their care plan could be more limited (e.g. "*When possible*, treatment recommendations accounted for current and previous treatment experiences and patient preferences") [97]. In other cases, patients participated in an initial 'contracting' visit with the collaborative care team to undergo assessment, receive education, discuss treatment options, and jointly develop a care plan that considered their treatment preferences [98, 99].

It was during the processes of goal setting and action planning that interprofessional collaboration was most apparent. During these steps, care managers partnered with patients, primary care physicians, psychiatrists, and potentially other professionals to jointly determine the goals of care and prepare a tailored action plan. Care plan development mostly focused on pharmacotherapy or psychotherapy treatment goals but in a few programs focused on or extended to self-management or wellness goals (such as in programs led by Simon [83],

Morgan [72] and Zimmerman [95]). The action planning process was frequently sequential, with an initial plan formulated by some team members (typically the care manager and mental health specialists) and then reviewed and validated by other team members (e.g. the primary care physician). Some reference to collaborative goal setting or action planning was identified in 47% and 69% of programs, respectively. In 82% of programs, the action plan was documented in a paper-based and/or electronic medical record format. In 8 programs, the care planning discussions were performed orally and there were no references to the documentation of the plan. Only 10 programs reported providing patients with a copy of the care plan.

The second phase of the care planning process involves the coordination of care and adjusting of the care plan. Processes related to coordinating and supporting care were described in 94% and 98% of programs featuring care planning, respectively. Care managers played an important role in coordinating patient care, including tests, treatments, self-management activities, team meetings, referrals, and follow-up supports. Supports to help patients achieve their goals were commonly provided by multiple members of the collaborative care team and took many forms, including monitoring and reinforcing progress, making treatment adjustments, promoting medication adherence, supporting self-management and problem solving, providing coaching and motivational supports, and helping patients connect with needed resources. In 20% of programs, supports also included the development of a formal relapse prevention plan. In 96% of programs, the care manager or collaborative care team reviewed patients' care plans and made adjustments needed to support patients' recovery, often following a stepped-care approach. Follow-up periods lasted from several weeks to 6 months in duration in 49% of programs, were between 6 and 12 months in 45% of programs, or lasted more than one year in 3 programs [56, 84, 100].

When involved, family members participated in the care planning process in several ways. In four programs [34, 43, 48, 96], family members were invited to participate in the development of the care plan by joining the patient in the initial session with the care manager. This provided an occasion for family members to receive education tailored to their needs and participate in care decisions. In two of these programs, family members were parents of youth receiving services from the collaborative care team [34, 43]. In three collaborative care programs that targeted low-income, predominantly Hispanic patients [51, 53, 96], family members were invited to participate in treatment and be sources of support for their loved ones. Family members also occasionally acted as interpreters that facilitated communication between patients and providers. In the IMPACT program targeting older adults with depression, clinicians found that enlisting the support of family members was "the most effective strategy in treating depressed older men" and that families were "an asset to their diagnosis and care plan" [101]. Family members were viewed as reliable sources of information, helped clinicians establish more accurate diagnoses, contributed to regular monitoring away from the clinic, and promoted compliance with the care plan.

## Shared decision making processes

When programs featured SDM, in ≈90% of cases a single decision–the choice of treatment (e.g. pharmacotherapy or psychotherapy, type of medication)–was the focus of the SDM process. Other decisions that were supported through SDM related to the choice of location of care, self-care options, and treatment duration or the frequency of contacts with the collaborative care team. The most commonly reported processes related to SDM included the exchange of information between patient and provider (65% of programs), the presentation of options to the patient (91% of programs), the exploration of patients' preferences (93% of programs), and the joint making of the decision by the patient and provider (42% of programs). Specific mention

of processes related to clarifying the problem/decision (9% of programs), discussing pros and cons of options (7% of programs), and revisiting decisions (28% of programs) were less frequent. None of the programs reported providing specific training to providers in SDM skills or included the use of decision aids to support the SDM process as part of their model of care.

We identified four evaluations specifically related to patients' involvement in decision-making. Clever and colleagues [102] examined the relationships between patient engagement in SDM and two outcomes: the receipt of guideline-concordant care and remission of depression. In their sample of 1706 adult patients with depression drawn from the Quality Improvement for Depression project (which combined data from several collaborative care programs [22, 103]), patients used a 5-point scale to self-report their involvement in decisions over the previous six months. Results showed that 45% of patients rated their involvement as either excellent or very good, 30% rated their involvement as good, and 25% rated it as fair or poor. Multivariate regression analyses revealed that patients with higher ratings of involvement in decision making were significantly more likely to receive guideline-concordant pharmacotherapy or psychotherapy over the 18-month follow-up period. Higher ratings of SDM were also associated with a significantly higher probability of remission from depression over the 18-month follow-up period. In both cases, findings remained significant after controlling for a range of confounders including the effects of collaborative care. In the studies led by Lin [97] and Gum [104], both teams evaluated whether the matching of treatment preferences to treatments received had an influence on depressive symptom improvement and other outcomes. Lin and colleagues conducted their study using data on 335 patients with depression involved in the collaborative care program led by Hedrick [57]. They found that 72% of patients received treatments that matched their initial preferences, and that this matching was positively associated with improvements in depression symptoms at 3-month follow-up. This effect remained after controlling for the effects of collaborative care, though results did not achieve significance at the 9-month follow-up assessment. Gum examined the predictors and impacts of preference-treatment matching in 1602 older patients with depression involved in the IMPACT program [23]. Receiving a preferred treatment was predicted by previous treatment experience, gender (men more likely to prefer medication, women more likely to prefer counseling), and depression severity (people with a more severe condition were more likely to prefer medication). Receiving a preferred treatment was not associated with improvements in depressive symptoms at 12 months, nor with other outcomes such as satisfaction with care. Finally, in the evaluation of the BRIDGE program [44], Cooper and colleagues examined the level of participation in decision making among a racially diverse sample of 132 adults with depression, finding that 68% rated their involvement as often or very often participatory (on a 5-point scale). The authors observed no changes in the likelihood of engaging in SDM between intervention and control groups over the 18-month follow-up period.

## Discussion

The purpose of this review was to describe practices related to personalized care planning and shared decision making in collaborative care programs for depressive and anxiety disorders. We performed an in-depth analysis of 150 collaborative care programs that drew from reports of clinical trials, quantitative and qualitative sub-studies, and study protocols. This is first synthesis to focus on the processes of personalized care planning and SDM within collaborative mental health care and among the first detailed investigations into the person-centeredness of these interventions.

The review sought to answer three main research questions. First, we were interested in knowing to what extent personalized care planning and SDM were included as strategies for

engaging patients and families in collaborative care. Our synthesis revealed that at least one of the strategies was included in 63 of the 150 collaborative care programs, or fewer than half of the programs. More specifically, 34% of programs described practices consistent with personalized care planning and 29% of programs featured practices consistent with SDM. This was a surprising result given the emphasis placed on supporting patient activation in several of the early, influential collaborative care programs [23, 105–107]. It is possible that, despite patient involvement being valued, it remained unclear to some program leaders how this idea could translate into concrete practices by members of the collaborative care team. Indeed, the conceptual foundations for many collaborative care programs can be traced to the Chronic Care Model developed in the late 1990s [108, 109]. This model stresses the importance of empowering and preparing patients to be more informed and active participants in their own care; however, the only engagement strategy emphasized by the model to achieve this (other than patient education) is support for self-management [109–111]. It is also possible that there may have been some reluctance on the part of program leaders to promote strategies such as personalized care planning and SDM within some programs. These strategies have not been not widely practiced in routine primary care and mental health services [14, 18, 20, 112, 113], largely due to various patient-related, provider-related, and system-related barriers [14, 18, 19, 114]. Among these barriers include a perceived lack of time to adopt these strategies, perceptions that patients lack the interest or capacity to engage in care planning or SDM, concerns about the clinical appropriateness of patients' choices, the limited availability of some treatment options, the threat to professional autonomy and culture of clinician authority, high staff workloads, and limited resources [14, 18, 19, 114]. Furthermore, while personalized care planning and SDM were promoted by institutions such as the Institute of Medicine [115] and have long been considered best practices on ethical grounds [116], the evidence base in support of them may not have been sufficiently robust until the past decade. As Katon and colleagues explained [117], involving patients in making choices about their treatments was still considered "controversial" in the early 2000s, largely due to uncertainties about the outcomes of this practice. These authors ultimately endorsed patient choice in their intervention because it more closely resembled 'real-world' clinical conditions. Still, the influence of various barriers may explain why fewer than half of programs featured strategies that are now considered fundamental to person-centered care [12, 118].

Our second research question concerned who participated in personalized care planning and SDM. We learned that in nearly three quarters of programs featuring personalized care planning, the process involved at least three members of the collaborative care team. This was most commonly the family physician, care manager, and psychiatrist, but occasionally other professionals including psychologists, nurses, pharmacists, social workers or other health specialists. The results demonstrate that an interprofessional approach to care planning is feasible in collaborative care. This approach is likely to be beneficial to both patients and providers, as it can help routinize collaborative discussions among team members, promote the integration of knowledge and expertise of each profession, strengthen the coordination and synchronisation of services, improve continuity of care, and support the development of more comprehensive care plans [119, 120]. Interestingly, while it was less common, in approximately half of programs that featured SDM an interprofessional approach to decision-making (IP-SDM) was also adopted. Here, IP-SDM primarily involved the family physician and care manager working collaboratively with the patient. As Chong and colleagues have noted, the IP-SDM approach can be challenging to implement in mental health care, especially when services are not integrated and communication between professionals is poor [121, 122]. However, collaborative care provides the ideal context for this practice, with our results highlighting the pivotal role of care managers in facilitating team-based decision-making. In several programs [41, 56,

79, 80], care managers met with patients to discuss treatment options and assess their preferences and then communicated these preferences to other members of the team (e.g. physicians, psychiatrists, psychologists, etc.), who formulated treatment recommendations that could be reviewed and accepted (or declined) by the patient and their physician. This asynchronous approach to decision-making is common when multiple professionals are involved [123] and can promote IP-SDM when professionals are not all co-located in the same clinical settings.

For our third research question, we wanted to know how and when personalized care planning and SDM were practiced within collaborative care. The synthesis allowed us to describe these strategies in detail but it also revealed several opportunities for quality improvement. For instance, fewer than half of programs described how patients were engaged in collaborative goal setting and most care plans were actually treatment plans that did not address a broader set of goals and activities that could promote patient recovery and wellness. In some health systems, care planning has evolved to become an increasingly bureaucratic exercise that is disconnected from patients' priorities and that fails to catalyze authentic teamwork [112, 113, 124]. To avoid this trap, collaborative care program leaders should ensure that care planning integrates goals and concerns formulated by the patient as well as joint action planning to achieve their preferred outcomes [13, 119]. Patients should be supported in formulating SMART (specific, measurable, assignable, realistic and time-related) goals as part of their care plan [119, 124], which not only personalizes their plan but also helps them develop a skill that is critical to their long-term self-management abilities [125]. Grundy and colleagues have also developed guidance for strengthening patient involvement in the care planning process (the 10Cs framework) that emphasizes a commitment to recovery-oriented plans that remain dynamic and responsive to change [124]. We also recommend that more attention be given to relapse prevention within care planning, as rates of relapse in depressive and anxiety disorders are high [126, 127] and preventing recurrence of episodes is critical to promoting better patient and population mental health [5]. With respect to SDM, most programs strove to provide patients with "choice" over their treatments but there is little evidence that providers helped patients understand the implications of those choices. Only three programs made reference to discussions that helped patients understand the key trade-offs between the benefits and harms of different treatment options. Previous studies of SDM in mental health care have similarly shown that this step is performed less often than other steps in the SDM process [128–131]. Yet, deliberations about the benefits, risks and potential harms of treatments is at the heart of SDM; it is a critical step in which clinical judgment, scientific evidence, and patients' values converge [132]. The absence of this step, and inconsistent practice of other steps such as clarifying and reviewing decisions, points to a need to place greater emphasis on SDM within the collaborative care model. This can be achieved by including training in SDM as part of the training providers receive when collaborative care is being implemented. There are now many examples of training programs in SDM that can inspire leaders of collaborative care programs [133]. We also recommend that programs make use of the growing number of patient decision aids that are designed to support SDM at the point-of-care. These tools can improve patients' involvement in decisions, their knowledge of options, their perceptions of benefits and harms, and their ability to make decisions congruent with their values [134].

Another important finding of our review was the minimal involvement of families in personalized care planning and SDM. Clearly, the involvement of family members in mental health care is a complex issue. While there is clear evidence that families often want to be active partners in care, they are often excluded from care planning and decision-making, leaving them to feel marginalized and disempowered [21, 135]. We did however identify several collaborative care programs that have seemingly overcome the barriers to family involvement. In particular, our findings suggest that families can be invited to attend initial sessions with the

care manager, during which they can share information, receive education, facilitate discussions, and participate in decisions and the development of the care plan. Hamann suggests that patients identify the family members that should be participants in care and that they be given the opportunity to clarify their desired role in care decisions [21]. When family members desire an active role, they should be given the opportunity to express their preferences on treatment options and how they may contribute to supporting their loved ones [21]. This process provides everyone with a foundation for sustained communication and partnership that can be critical to the second broad phase of care planning involving the coordination of care and responsive adjusting of the patient's care plan [135]. Such a foundation of communication can also help collaborative care teams detect and address any needs for support that may arise over time from family members themselves.

## Limitations to the review

This study has some limitations. First, our literature searches captured collaborative care programs that were evaluated in clinical trials and as such we may have missed other innovative or more person-centered programs that had not been evaluated with trial methods. Second, while our criteria for identifying personalized care planning and SDM processes were comprehensive, detailed descriptions of these strategies were not always provided and we cannot be certain that our findings reflect the realities of clinical practice within these programs. It is possible that our findings underestimate the extent to which these engagement strategies were practiced in real-world practice settings, as some may consider shared care planning and decision-making to be already ingrained in practice and thus not noteworthy as collaborative care components. However, given the absence of training and tools available to support these practices we may assume that there could be considerable variations in practice with respect to care planning and SDM within the programs examined. It would have been beneficial to contact study authors and program leaders to gather additional information on these strategies in order to increase the confidence in our findings.

## Limitations to the evidence

This review used multiple strategies to collect as much information and data as possible on personalized care planning and SDM processes within collaborative care programs. However, these two components of collaborative care remain inconsistently described across studies. Improved reporting of these strategies for engaging patients and families in their care would allow for a stronger appreciation of the person-centeredness of programs. The conceptualizations used within this review could serve as a foundation for improved reporting in future collaborative care studies.

## Conclusion

Personalized care planning and shared decision making are two evidence-based strategies for improving the involvement of patients and families in collaborative mental health care. Both strategies are core clinical practices for modern mental health and primary care services that are person-centered and recovery-oriented. However, this systematic review suggests that personalized care planning and SDM have not been fully integrated within the collaborative care model and that these strategies are either absent or sub-optimally practiced in many programs for common mental disorders. Given the importance of the collaborative care model for improving primary mental health services worldwide, this represents a major missed opportunity to spread and scale-up collaborative practices in which patients and families are full partners in care. Strengthening practices related to personalized care planning and SDM would

likely enhance the overall effectiveness of collaborative care and these programs' impact on the mental health of populations.

## Supporting information

**S1 File. CINAHL search.**
(DOCX)

**S2 File. Examples of personalized care planning and shared decision making processes.**
(DOCX)

**S3 File. PRISMA checklist.**
(DOCX)

## Acknowledgments

The authors wish to thank Élodie Chenard, Thierry Provencher, and Jessica Hébert for their assistance with the study selection process. We are also grateful to Drs France Légaré, Emmanuelle Careau, Maud-Christine Chouinard, André Delorme, Maman Joyce Dogba, Janie Houle, Nick Kates, Sarah Knowledge, Donald Nease Jr, and Hervé Zomahoun, as well as Ms. Neasa Martin and members of the Association Québécoise pour la readaptation psychosociale for their contributions to the initial stages of this study.

## Author Contributions

**Conceptualization:** Matthew Menear, Ariane Girard, Michèle Dugas, Michel Gervais, Michel Gilbert, Marie-Pierre Gagnon.

**Data curation:** Matthew Menear, Michèle Dugas.

**Formal analysis:** Matthew Menear, Ariane Girard, Michèle Dugas, Michel Gervais, Michel Gilbert.

**Funding acquisition:** Matthew Menear, Michel Gervais, Michel Gilbert, Marie-Pierre Gagnon.

**Investigation:** Matthew Menear.

**Methodology:** Matthew Menear, Michèle Dugas.

**Project administration:** Matthew Menear.

**Supervision:** Matthew Menear.

**Validation:** Matthew Menear, Ariane Girard, Michèle Dugas.

**Writing – original draft:** Matthew Menear, Ariane Girard, Michèle Dugas, Michel Gervais, Michel Gilbert, Marie-Pierre Gagnon.

**Writing – review & editing:** Matthew Menear, Ariane Girard, Michèle Dugas, Michel Gervais, Michel Gilbert, Marie-Pierre Gagnon.

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
