## [Decision Letter · Decision Letter 0]

6 Nov 2021

PONE-D-21-13986Personalized care planning and shared decision making in collaborative care programs for depression and anxiety disorders: a systematic reviewPLOS ONE

Dear Dr. Mennear,

Thank you for submitting your manuscript to PLOS ONE. After careful consideration, we feel that it has merit but does not fully meet PLOS ONE’s publication criteria as it currently stands. Therefore, we invite you to submit a revised version of the manuscript that addresses the points raised during the review process.

Please see comments of the reviewer which require revision.  

Please submit your revised manuscript by November 30, 2021.  If you will need more time than this to complete your revisions, please reply to this message or contact the journal office at plosone@plos.org. Please include the following items when submitting your revised manuscript:A rebuttal letter that responds to each point raised by the academic editor and reviewer(s). You should upload this letter as a separate file labeled 'Response to Reviewers'.A marked-up copy of your manuscript that highlights changes made to the original version. You should upload this as a separate file labeled 'Revised Manuscript with Track Changes'.An unmarked version of your revised paper without tracked changes. You should upload this as a separate file labeled 'Manuscript'.If applicable, we recommend that you deposit your laboratory protocols in protocols.io to enhance the reproducibility of your results. Protocols.io assigns your protocol its own identifier (DOI) so that it can be cited independently in the future. For instructions see: https://journals.plos.org/plosone/s/submission-guidelines#loc-laboratory-protocols. Additionally, PLOS ONE offers an option for publishing peer-reviewed Lab Protocol articles, which describe protocols hosted on protocols.io. Read more information on sharing protocols at https://plos.org/protocols?utm_medium=editorial-email&utm_source=authorletters&utm_campaign=protocols.

We look forward to receiving your revised manuscript.

Kind regards,

Gerard Hutchinson, MD

Academic Editor

PLOS ONE

Journal Requirements:

Reviewers' comments:

Reviewer's Responses to Questions

**Comments to the Author**

1. Is the manuscript technically sound, and do the data support the conclusions?

Reviewer #1: Yes

2. Has the statistical analysis been performed appropriately and rigorously? 

Reviewer #1: N/A

3. Have the authors made all data underlying the findings in their manuscript fully available?

Reviewer #1: Yes

4. Is the manuscript presented in an intelligible fashion and written in standard English?

Reviewer #1: Yes

5. Review Comments to the Author

Reviewer #1: This is a well conducted and well written narrative review of the use of personalised care planning and shared decision making in collaborative care programmes. It follows similar articles by the authors on strategies used to engage patients and families in collaborative care programmes. I have no qualms in recommending this article be accepted for publication as written.

Strengths of the article include:

- well-defined research questions (p6)

- pre-registration with the PROPSPERO (p7)

- application of the PRISMA guidelines (p7)

- retrieval of 'sibling' reports as well as academic trial reports (p7)

- use of a previously validated search strategy, that is well described in other peer-reviewed publications by the authors (p7)

- a well-considered definition of the collaborative care model, as it can mean different things to different people (p8)

- well-defined and validated definitions of personalised care planning and shared decision-making (pp9-10)

- the discussion makes some well-reasoned and valid points based on the methodology and results

My only minor comments that the authors may wish to consider are:

1. The authors explain the emphasis placed on personalised care planning and SDM by developers and innovators of the collaborative acre approach, but they could explain WHY the inclusion of these elements was considered necessary by the original proponents of the CCM. They do explain this later (pp24-25) e.g. correlation between SDM and receipt of evidence-based treatment, reduction in depressive symptoms, and so on. They may want to introduce these impacts in the Introduction as well.

2. While the authors have extracted studies that expressly mention personalised care planning or SDM as part of the collaborative care offer, it may be of course that SDM occurred naturally in real world settings throughout collaborative care programmes (for example in discussions with patient and carers on whether to start antidepressants, or what therapy to consider) but was not made an explicit aim of a study or described specifically because SDM can be considered the default position of some professionals e.g. primary care physicians. The authors may want to reflect upon this, in that they may have underestimated the degree to which SDM actually occurs in real-world settings.

3. On p22 when the authors write 'Processes related to coordinating and supporting care were described

in 94% and 98% of programs, respectively' - do they mean 94-98% of programmes that undertook care planning (not of all 150 collaborative care programmes that they included)?

6. PLOS authors have the option to publish the peer review history of their article (what does this mean?). If published, this will include your full peer review and any attached files.

Reviewer #1: **Yes: **Parashar Pravin Ramanuj

---

## [Author Response · Author response to Decision Letter 0]

2 Apr 2022

Response to Review (responses in Calibri font)

Academic Editor Comments:

Response: We have made some minor modifications to the manuscript’s title page to ensure that we have met the journal’s style requirements.

2. In your Data Availability statement, you have not specified where the minimal data set underlying the results described in your manuscript can be found.

Response: The minimal data set for this study has been published in the Harvard Dataverse and is available using the following URL: https://doi.org/10.7910/DVN/MWZOIE.

3. Please review your reference list to ensure that it is complete and correct.

Response: Our reference list has been reviewed and is complete.

Reviewer Comments:

1. The authors explain the emphasis placed on personalised care planning and SDM by developers and innovators of the collaborative acre approach, but they could explain WHY the inclusion of these elements was considered necessary by the original proponents of the CCM. They do explain this later (pp24-25) e.g. correlation between SDM and receipt of evidence-based treatment, reduction in depressive symptoms, and so on. They may want to introduce these impacts in the Introduction as well.

Response: We thank the reviewer for this suggestion. Early papers on collaborative care did not always explain why it was important to engage and activate patients but we know that their models were largely influenced by the emergence of other frameworks like the Chronic Care Model. Katon in his work in the early 2000s provided more context in this trend by referring to the need for actions outside the clinical context and the importance of taking into consideration patient preferences to promote adherence to treatments. We have thus made some modifications to our introduction to better explain why mental health service providers became interested in engaging patients in their mental health care.

Our introduction now states the following:

(Page 4) Views on the roles of patients and families as active partners in collaborative care have similarly evolved (8, 9). There has been a growing acknowledgement that much of the work to promote patient recovery happens outside the clinic, and involves patients and families becoming better informed about their conditions, taking greater responsibility in illness management and learning to adopt psychosocial and lifestyle changes that can improve their well-being (9). Today, some view patients and families as the most important members of the collaborative care team and emphasize the need for supports in helping them become partners in care (10).

(Page 5) People with common mental disorders may encounter a number of decisions over the course of their care that are preference-sensitive and that would warrant an SDM approach (18, 20). They may also hold strong preferences for some options over others, such as the types of treatments they wish to receive.

2. While the authors have extracted studies that expressly mention personalised care planning or SDM as part of the collaborative care offer, it may be of course that SDM occurred naturally in real world settings throughout collaborative care programmes (for example in discussions with patient and carers on whether to start antidepressants, or what therapy to consider) but was not made an explicit aim of a study or described specifically because SDM can be considered the default position of some professionals e.g. primary care physicians. The authors may want to reflect upon this, in that they may have underestimated the degree to which SDM actually occurs in real-world settings.

Response: We definitely agree that it is possible that some personalized care planning and shared decision making was occurring within these programs and simply not reported or emphasized as a core component of their collaborative care model. We had acknowledged this possibility in our “Limitations to the review” section but have made some further modifications to ensure that this possibility stands out more clearly. Also, while we acknowledge this possibility, we also know from the literature that while clinicians may think that they are practicing these engagement strategies, more rigorous assessments of their practice show that they are often not doing so optimally and that without training and tools there are likely to be wide variations in these practices. We now state the following:

(Page 32) It is possible that our findings underestimate the extent to which these engagement strategies were practiced in real-world practice settings, as some may consider shared care planning and decision-making to be already ingrained in practice and thus not noteworthy as collaborative care components. However, given the absence of training and tools available to support these practices we may assume that there could be considerable variations in practice with respect to care planning and SDM within the programs examined.

3. On p22 when the authors write 'Processes related to coordinating and supporting care were described in 94% and 98% of programs, respectively' - do they mean 94-98% of programmes that undertook care planning (not of all 150 collaborative care programmes that they included)?

Response: Yes, we mean 94-98% of programs that featured care planning. This is now specified, both in that paragraph and in early paragraphs where necessary.

---

## [Editor Report · Decision Letter 1]

4 May 2022

Personalized care planning and shared decision making in collaborative care programs for depression and anxiety disorders: a systematic review

PONE-D-21-13986R1

Dear Dr.Mennear,

We’re pleased to inform you that your manuscript has been judged scientifically suitable for publication and will be formally accepted for publication once it meets all outstanding technical requirements.

Kind regards,

Gerard Hutchinson, MD

Academic Editor

PLOS ONE
---

## [Editor Report · Acceptance letter]

2 Jun 2022

PONE-D-21-13986R1 

Personalized care planning and shared decision making in collaborative care programs for depression and anxiety disorders: a systematic review 

Dear Dr. Menear:

I'm pleased to inform you that your manuscript has been deemed suitable for publication in PLOS ONE. Congratulations! Your manuscript is now with our production department. 

Kind regards, 

on behalf of

Dr. Gerard Hutchinson 

Academic Editor

PLOS ONE